# Validity of acute cardiovascular outcome diagnoses in European electronic health records: a systematic review protocol

Jennifer Anne Davidson,[1] Amitava Banerjee [iD],[2] Rutendo Muzambi [iD],[1] Liam Smeeth,[1] Charlotte Warren-Gash[1]

[1]Faculty of Epidemiology & Population Health, London School of Hygiene and Tropical Medicine, London, UK
[2]Farr Institute of Health Informatics Research, University College London, London, UK

**Correspondence to**
Jennifer Anne Davidson;
jennifer.davidson@lshtm.ac.uk

## ABSTRACT

**Introduction** Cardiovascular diseases (CVDs) are among the leading causes of death globally. Electronic health records (EHRs) provide a rich data source for research on CVD risk factors, treatments and outcomes. Researchers must be confident in the validity of diagnoses in EHRs, particularly when diagnosis definitions and use of EHRs change over time. Our systematic review provides an up-to-date appraisal of the validity of stroke, acute coronary syndrome (ACS) and heart failure (HF) diagnoses in European primary and secondary care EHRs.

**Methods and analysis** We will systematically review the published and grey literature to identify studies validating diagnoses of stroke, ACS and HF in European EHRs. MEDLINE, EMBASE, SCOPUS, Web of Science, Cochrane Library, OpenGrey and EThOS will be searched from the dates of inception to April 2019. A prespecified search strategy of subject headings and free-text terms in the title and abstract will be used. Two reviewers will independently screen titles and abstracts to identify eligible studies, followed by full-text review. We require studies to compare clinical codes with a suitable reference standard. Additionally, at least one validation measure (sensitivity, specificity, positive predictive value or negative predictive value) or raw data, for the calculation of a validation measure, is necessary. We will then extract data from the eligible studies using standardised tables and assess risk of bias in individual studies using the Quality Assessment of Diagnostic Accuracy Studies 2 tool. Data will be synthesised into a narrative format and heterogeneity assessed. Meta-analysis will be considered when a sufficient number of homogeneous studies are available. The overall quality of evidence will be assessed using the Grading of Recommendations, Assessment, Development and Evaluation tool.

**Ethics and dissemination** This is a systematic review, so it does not require ethical approval. Our results will be submitted for peer-review publication.

**PROSPERO registration number** CRD42019123898

## Strengths and limitations of this study

► This systematic review will comprehensively evaluate the validity of selected major cardiovascular diagnoses (stroke, acute coronary syndrome and heart failure) in electronic health record (EHR) databases used in the provision of primary and secondary clinical care in Europe by searching five bibliographic databases and two grey literature sources with no language or date restrictions.

► The Preferred Reporting Items for Systematic Reviews and Meta-Analyses statement will be followed ensuring this systematic review provides high-quality scientific results.

► There may be heterogeneity in the results produced by our systematic review due to differences in EHR design and use between countries, in particular the relevance of our findings to countries outside of Europe requires further evaluation.

contribute to morbidity and mortality. Ischaemic heart disease followed by stroke have been the global leading causes of death for 15 years, and in 2016 accounted for 15.2 million deaths.[1] Also in 2016, worldwide more than 13 million people were estimated to have suffered a stroke[2] with healthcare expenditure on stroke estimated to be 3%–5%.[3–5] An estimated 26 million people are living with HF,[6] a chronic condition with acute episodes. HF is estimated to account for 1%–2% of healthcare expenditure in Europe and the USA.[7] Added to the complication of estimating the burden of CV conditions is changes to definitions; the fourth universal definition of myocardial infarction (MI) was issued in 2018.[8]

Increases in the incidence and prevalence of CV conditions are in part due to an ageing population,[9] but also due to modifiable risk factors, such as smoking, unhealthful diet and lack of physical exercise, and non-modifiable risk factors, such as sex and ethnicity.[10] A

## INTRODUCTION
### Rationale

Stroke, acute coronary syndrome (ACS) and heart failure (HF) are the three cardiovascular (CV) conditions, which substantially

range of factors, such as pollution, infections, emotional stress and physical exertion, can also trigger acute CV events particularly in those with pre-existing cardiovascular disease (CVD).[11–13]

Electronic health record (EHR) databases are derived from clinical care records and contain longitudinal patient data on diagnoses, treatment and other clinically relevant variables, such as smoking. Administrative databases were developed for financial and management purposes to allocate funding or billing of insurance claims. While both are types of computerised health-related data that have been widely used for research, they are quite distinct. In particular, the completeness and accuracy of the morbidity data may differ in the two types of data because of the very different reasons why the data were recorded in the first place. In settings where both clinical and administrative data are available, results from some studies suggest the quality of administrative data is lower.[14 15]

High-quality EHR-based research depends on correct classification of cases and non-cases. Several systematic reviews have previously appraised the validation of specific European EHRs[16–18] as well as specific conditions recorded within EHRs, including CVD.[19–23] The previous systematic reviews on the validity of CVD diagnoses included EHRs, along with administrative databases and vital registration databases. McCormick et al reported that the positive predictive value (PPV) of stroke diagnosis ranged from 32% to 98%, with the majority of included studies using administrative data from North America,[22] while Woodfield et al identified PPVs of >70% for stroke based on the results that included a greater proportion of studies from Europe,[20] where EHRs are widely used. McCormick et al's review of HF diagnosis validity obtained PPVs ranging from 17% to 100% but only contained four studies outside of North America.[23]

### Aim and objectives

The aim of our systematic review is to provide an up-to-date appraisal of the validity of stroke (and its subtypes), ACS (including MI and other ACS) and HF diagnoses in adults focused on European EHRs used in primary and secondary care. Our objectives are to:
1. Summarise and pool estimates of the sensitivity, specificity, PPV and negative predictive value (NPV) of stroke, ACS and HF diagnoses compared with a suitable reference standard.
2. Determine whether estimates differ by study population, validation method, data source, diagnosis and time period.

### METHODS

This protocol has been prepared using the Preferred Reporting Items for Systematic Reviews and Meta-Analyses Protocols guidelines.[24]

### Eligibility criteria

We used the PICOS (Population, Intervention, Comparator, Outcomes and Study design) framework to formulate the research question and eligibility criteria for our review, but adapted this to replace 'Intervention' with 'Index test', the modification recommended for systematic reviews of diagnostic test accuracy.[25] This modification was chosen as it is the closest resemblance to the validation of EHRs.

### Population

Eligible studies will include records of adults aged 16 years or older from any European primary or secondary care national or the regional EHR database. We will exclude studies that validate administrative (insurance claims or billing) databases, disease registries or vital registration systems, as well as studies that validate locally held databases. EHRs and administrative databases collect data for different purposes and may differ in accuracy, we are interested in the validity of EHRs used in clinical settings. The comprehensive data capture methods used to populate disease registries mean that these datasets are often used as the gold standard in validation of EHRs so would be unsuitable to include in our validation estimates. Vital registration systems only capture deaths so unless combined with EHRs, the data is not by itself useful in non-mortality-related research. Finally, data from locally held databases are unlikely to be captured in centralised EHRs used in research and therefore validation results are not informative for researchers.

### Index test

We are interested in records with clinical codes, for example, International Classification of Primary Care (ICPC) or International Classifications of Disease (ICD), which identify a diagnosis of stroke (and its subtypes), ACS (including MI and other ACS) or HF in primary or secondary care EHRs. The ICD-9 and ICD-10 codes, we assume studies, will include (and which we look to validate) in their stroke, ACS and HF definitions are presented in table 1.

### Comparator

To be included, studies must have validated against an internal or external reference standard. Eligible external reference standards include manual review of medical records, patient or clinical questionnaire, or comparison with an independent second database. Internal within database comparison includes validation against a

**Table 1** Provisional list of ICD-9 and ICD-10 codes included in diagnoses of interest

| Diagnosis | ICD-9 | ICD-10 |
|---|---|---|
| Stroke | 430, 431, 432, 433, 434 | I60, I61, I62, I63, I64 |
| Acute coronary syndrome | 410, 411 | I20.0, I21, I22, I24, I49 |
| Heart failure | 428 | I11.0, I13.0, I13.2, I50 |

ICD, International Classifications of Disease.

diagnosis algorithm or comparison of clinical codes with anonymised free text.

## Outcome

Studies must either report (1) at least one of the following validation estimates; sensitivity, specificity, PPV and NPV or (2) data which allows at least validation estimates to be calculated.

## Study design

We will include any type of study from any time period published in any language that includes the validation of the recording of stroke, ACS or HF diagnoses in an EHR database, regardless of if this was the main objective of the study.

## Information sources

To review published and in-process citations the following databases will be searched from inception to April 2019; MEDLINE, EMBASE, SCOPUS, Web of Science and Cochrane Library. Using OpenGrey and EThOS, we will search for the relevant grey literature. Bibliographies of national EHR databases used for research will also be searched.

## Search strategy

The search strategy will include subject heading terms and free text (title and abstract) for the concept of acute CV events using the synonyms of stroke, ACS and HF as well as the concepts of EHRs and validation. We will limit our search to studies conducted using European EHRs. Provisional search terms have been developed for MEDLINE (online supplementary appendix 1), and once finalised will be transcribed into corresponding searches for the other aforementioned information sources. We will also review the reference list of other relevant systematic reviews identified during the screening process as well as of articles included in our review to identify further potentially relevant studies.

## Study records

### Data management

Citations from the searched databases will be exported into Endnote X9. Electronic deduplication of records will be conducted, followed by manual deduplication where necessary.

### Selection process

For the initial screening stage, two authors (JAD and RM) will independently review all titles and abstracts to assess whether they fulfil the eligibility criteria for inclusion. To reduce the risk of missing potentially relevant studies, reviewers will adopt a lenient approach for this first level of screening including any study that validate stroke, ACS or HF diagnoses in EHRs. Full-text articles for studies that meet the review criteria will be obtained and reviewed by the two authors (JAD and RM). The reasons for rejection of articles during the full-text screening process will be noted according to a hierarchical list: (1) could not

obtain full text, (2) did not conduct validation, (3) duplicate study, (4) wrong outcome, (5) wrong index (ie, not a primary or secondary care EHR in Europe), (6) not a suitable comparator or (7) no validation estimate or insufficient data to calculate. Any discrepancies at either the initial screening or full-text screening will be discussed by the two reviewers, with a third author (CW-G) consulted when necessary.

### Data collection process

To extract information for each study selected for final inclusion, data extraction tables will be piloted by the two authors (JAD and RM) for three studies with changes made, if required. We will then dual extract data from a further 10% of studies using the finalised template. If there are any significant discrepancies between the two reviewers, then we will conduct parallel data extraction for a further 10% of studies, again checking for discrepancies. This process will be repeated until no further discrepancies occur, at which stage the remaining data extraction will be completed by the single reviewer (JAD). At each stage, the third author (CW-G) will be consulted when the two reviewers cannot resolve discrepancies.

## Data items

Similar to our search strategy, we will use the PICOS framework to systematise the extraction of data from each study. We will use a standardised template containing information on each of the following five domains:

1. Population: participants, age and sex, inclusion and exclusion criteria.
2. Index test: EHR country, EHR name, EHR setting (primary or secondary care), EHR coding system (ICPC, ICD, etc), EHR coverage (regional or national), diagnoses validated including whether incident or prevalent, specific diagnoses codes validated.
3. Comparator: method of validation, description of method.
4. Outcome: number of participant diagnoses planned, number of diagnoses conducted, measures of validity and raw data to calculate measures of validity.
5. Study characteristics: authors, publication year, language, study design, study period, main aim of the study (validation or not validation).

## Outcomes and prioritisation

The outcome is any validation estimate of stroke (including all subtypes), ACS (MI or other ACS) or HF. The study has no secondary outcomes.

## Risk of bias in individual studies

To assess bias, we will use a tailored version of the Quality Assessment of Diagnostic Accuracy Studies 2 (QUADAS-2) tool, which is used for assessing diagnostic accuracy studies,[26] based on the previous modifications made for assessing the validity of diagnostic coding in EHRs.[20 21] We will consider bias in each of the domains included in QUADAS-2; patient selection, index test, reference standard and flow and timing. In the context of our review,

index test translates to the clinical codes validated. We will produce a summary risk of bias figure, as well as an additional table explaining each judgement.

Two authors (JAD and RM) will independently pilot the tailored QUADAS-2 tool, assessing bias in three of the included studies. Any necessary changes will be made to the tool and dual assessment by the two reviewers will be done with the finalised tool for a further 10% of studies. If there are significant discrepancies, we will continue parallel risk of bias assessment for another 10% of studies, repeating the process until no further discrepancies occur. Assessment of the remaining studies will be completed by the single reviewer (JAD). At each stage, the third author (CW-G) will be consulted when the two reviewers cannot resolve discrepancies

### Data synthesis and metabias(es)

We will describe key study characteristics and use a narrative synthesis and forest plots to summarise the validity of each of stroke, ACS and HF diagnoses in European primary and secondary care EHRs. The $I^2$ statistic will our guide judgements about the level of statistical heterogeneity between the studies. We will use the Cochrane's suggested guide to grade the heterogeneity as a low (0%–40%), moderate (30%–60%), substantial (50%–90%) or considerable (75%–100%) obtained from the $I^2$ statistic.[27] If there is the sufficient number of studies selected, we will explore the reasons for heterogeneity. We will compare heterogeneity before and after removing the studies that deemed to be at a high risk of bias overall and by subgroups of: (1) study populations, that is, specific demographic or clinical groups, (2) validation method, (3) data source, that is, primary care and secondary care EHRs, (4) specific diagnosis, that is, incident or prevalent and stroke or ACS subtype and (5) variation in validity estimates over time.

We will consider conducting meta-analyses for each CV condition to calculate pooled effect estimates for sensitivity, specificity, PPV and NPV if studies are sufficiently homogeneous. Meta-analyses would be conducted by the aforementioned subgroups. Our choice of a fixed or random effects model would also be guided by the level of heterogeneity, with random effects meta-analysis methods followed if there is substantial heterogeneity.

### Confidence in cumulative evidence

Two reviewers (JAD and RM) will independently use the Grading of Recommendations, Assessment, Development and Evaluation (GRADE) tool[28] to judge the certainty of cross-study evidence for the validity of diagnoses in EHRs and their use in research. Any discrepancies between the two reviewers' judgements will be discussed and resolved, if necessary consulting the third author (CW-G). We will examine stroke, MI and HF diagnoses in EHRs for; overall risks of bias, inconsistency, indirectness, imprecision and publication bias with the production of funnel plots. The strength of evidence will be categorised as

high, moderate, low and very low. Our judgements will be presented in a summary of findings table.

### Patient and public involvement

Patients and/or public were not involved in this systematic review.

## DISCUSSION

This systematic review will provide an up-to-date assessment of the validity of primary and secondary care EHRs used for stroke, ACS and HF research. To our knowledge, this will be the first systematic review to focus solely on the validity of CVD diagnoses in EHRs. Previous systematic reviews have included EHRs along with administrative databases and vital registration databases. Each of these data sources has a different primary purpose, which in turn will impact the validity of the systems. One previous systematic review of MI diagnoses identified the accuracy for vital registration databases was lower (all PPVs≤59%) than the hospitalisation data (three-quarters of studies PPV >59%).[21] However, hospitalisation grouped EHRs and administrative databases together, so it is unclear if the one-quarter of studies with a PPV ≤59% differed by data source to those with higher PPVs.

Our systematic review will also serve to update several aspects covered by the previous systematic reviews validating CVD diagnoses. McCormick et al's 2010 review of MI diagnoses only identified three studies that validated the ICD-10 coding.[21] while Rubbo et al's 2014 review identified eight studies.[19] The majority of European countries implemented ICD-10 in the late 1990s. Our search run in April 2019 aims to identify more recent studies validating ICD-10 CVD diagnoses, the results of which are most relevant to today's research. In the majority of studies included in the previous systematic review of HF diagnoses, conducted in 2010, sensitivity was <69%.[23] Only one of the three included European studies reported sensitivity, this was 43%.[29] With an increase in the prevalence of HF,[6] and therefore accompanying public health research, we hypothesise that more studies validating HF diagnoses in EHRs will have been published between 2010 and 2019, the results of which will inform current HF research. Similarly, previous systematic reviews validating stroke diagnoses identified variation in accuracy by stroke subtype,[20 22] with the inclusion of up-to-date studies, we aim to analyse temporal changes in validity estimates with the assumption that more recent studies should have higher and more consistent estimates across stroke subtype. We also aim to present results for ACS other than MI, such as unstable angina which have not been included in any previous systematic review.

Our systematic review benefits from searching multiple databases with no language barriers, compared with the previous systematic reviews of CVD diagnoses, which either only searched 1–2 databases or only included English language studies. There are some limitations to our systematic review. First, by aiming to validate EHRs,

rather than broader computerised health-related data-sets, we have limited our review to Europe where EHRs operate nationally or covering nationally representative populations with widespread use in research. Consequently, our findings will not be applicable to administrative databases, also commonly used in research. Second, our results will not necessarily be applicable to countries outside of Europe using EHRs, if the design and utility of the EHRs differ. Lastly, previous systematic reviews on CVD diagnoses have been unable to conduct meta-analyses due to the level of heterogeneity identified in their results. We hope that by limiting our systematic review to EHRs in Europe, many of which are set up and operated in similar ways, the level of heterogeneity between the studies will be reduced. However, we will still be limited by variation in the reference standard used and differences in the codes included in validation. Therefore, it may not be possible to conduct any meta-analysis.

Overall, our systematic review should provide useful and up-to-date findings to inform researchers on the validity of using EHRs in their research.

## ETHICS AND DISSEMINATION

Important protocol amendments will be documented and a justification for deviating from the original protocol provided in a protocol addendum. The findings of this review will be submitted to a peer-reviewed journal.

**Contributors** CW-G conceived the study idea. JAD led the design of the study with the contributions from AB, LS and CW-G. JAD drafted the methods and analysis and revised the protocol following authors' comments from CW-G, AB, RM and LS. All authors approved the final version of the protocol.

**Funding** JAD was supported by a British Heart Foundation PhD Studentship (FS/18/71/33938).

**Disclaimer** The funders have no input on the protocol development and will not have influence on the conduct, analysis, interpretation or publication of the study results.

**Competing interests** None declared.

**Patient consent for publication** Not required.

**Ethics approval** Ethical review is not required as this study is a systematic review.

**Provenance and peer review** Not commissioned; externally peer reviewed.

**ORCID iDs**
Amitava Banerjee http://orcid.org/0000-0001-8741-3411
Rutendo Muzambi http://orcid.org/0000-0003-0732-131X

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
