## [Reviewer comments · BMJ Open]

ARTICLE DETAILS

TITLE (PROVISIONAL)	The validity of acute cardiovascular outcome diagnoses in European electronic health records: a systematic review protocol
AUTHORS	Davidson, Jennifer; Banerjee, Amitava; Muzambi, Rutendo; Smeeth, Liam; Warren-Gash, Charlotte

VERSION 1 - REVIEW

REVIEWER	Niamh Merriman Royal College of Surgeons in Ireland
REVIEW RETURNED	16-May-2019

GENERAL COMMENTS	Synopsis: The authors present a protocol for a systematic review of the validity of cardiovascular diagnoses in European electronic health records. Introduction: The authors seem to be making the point that inclusion of administrative health data results in a less precise estimate and that is why they should not include studies that use claims/billing data. Is this why they have limited their search to Europe? I would suggest making this point more explicitly, with similar justification for not including studies based on disease registry data (since this is also an exclusion criterion). Methods: The authors list a broad range of cardio/cerebrovascular events for inclusion. Taking stroke as an example, will the authors include all main stroke pathologies and associated ICD codes (ischaemic, intracranial haemorrhage, subarachnoid haemorrhage)? What about TIA? I think a table of acceptable ICD codes for each event (stroke, acute coronary syndrome, heart failure) would be useful for the reader. Selection process: Be explicit as to whether independent double screening of full text studies will take place. Data extraction: I would strongly recommend that that more than one person extract data from every study to minimise errors and reduce potential biases being introduced by review authors, as per the Cochrane Handbook. Risk of Bias: Again, two reviewers should independently assess risk of bias in every included study. GRADE quality appraisal: Again, two reviewers should independently assess study quality.
--

	Discussion: The discussion section is 4 lines long and needs revision. There is no discussion of the limitations or challenges of conducting this type of systematic review. Perhaps the authors could recap on what previous reviews have found and state how they plan for theirs to be different and what specifically it can add to the evidence base. Minor points: Page 4, line 14: The authors' last point is speculative as they have not conducted the review yet. Suggest removing. Page 4, line 53: "...depends upon the ability of accurately identifying..." awkward sentence, suggest rewording. The ability of whom/what? Page 7, line 40: suggest including study quality as a subgroup analysis for heterogeneity.
--	---

REVIEWER	Changyong Feng Department of Biostatistics and Computational Biology University of Rochester Rochester, NY, USA
REVIEW RETURNED	29-May-2019

GENERAL COMMENTS	(1) For a small meta-analysis like this one, I'm not sure they need to publish a research protocol in BMJ. It seems a protocol for internal review is good enough. (2) They don't have clear statistical analysis plan for the outcomes in the protocol.
--

VERSION 1 – AUTHOR RESPONSE

Reviewer: 1

C2. Introduction: The authors seem to be making the point that inclusion of administrative health data results in a less precise estimate and that is why they should not include studies that use claims/billing data. Is this why they have limited their search to Europe? I would suggest making this point more explicitly, with similar justification for not including studies based on disease registry data (since this is also an exclusion criterion).

R2. We have updated our Introduction (paragraph 3) to explain the distinction between clinical EHRs and administrative research datasets (page 3, lines 42-50). It is plausible that administrative datasets provide less precise estimates than clinical EHRs due to the different purposes of the systems. There is little evidence to support this given there are few settings where claims datasets and EHRs are both widely used. However, we have included what references we can to support this point (page 3, line 50). In the next section of the Introduction (paragraph 4) we have expanded the text to state that EHRs are widely used in Europe compared to administrative data elsewhere (page 4, line 6). We hope this explains why we are limiting our review to Europe. Finally, in the 'Eligibility criteria' of the Methods after stating which types of database we are excluding we have given a rationale for each of the exclusions (page 4, lines 40-48).

C3. Methods: The authors list a broad range of cardio/cerebrovascular events for inclusion. Taking stroke as an example, will the authors include all main stroke pathologies and associated ICD codes (ischaemic, intracranial haemorrhage, subarachnoid haemorrhage)? What about TIA? I think a table of acceptable ICD codes for each event (stroke, acute coronary syndrome, heart failure) would be useful for the reader.

R3. All subtypes of stroke are included in our definition. We have clarified this in the 'Aim and objectives' section as well as in the 'Eligibility criteria' of the Methods by stating "stroke (all subtypes)".

We have not included TIA in our definition. TIA can be a difficult diagnosis to make clinically – it is hard to recognise so may be under ascertained and can be vague and transitory so likely to have low specificity. Given these issues, our prior expectation is that TIA will be badly recorded. Additionally, it is rarely used as a major vascular outcome in research.

Thank you for your helpful suggestion of a table with ICD codes, we have added a table of provisional codes (Table 1).

C4. Selection process: Be explicit as to whether independent double screening of full text studies will take place.

R4. We have now clarified this by adding "and reviewed by the two authors (JD and RM)" to the end of the sentence on full text review.

C5. Data extraction: I would strongly recommend that that more than one person extract data from every study to minimise errors and reduce potential biases being introduced by review authors, as per the Cochrane Handbook.

R5. R5. Thank you for this recommendation. We agree that ideally two reviewers should independently conduct data extraction for all studies. Our systematic review is being conducted as part of a PhD and our proposal to dual extract data for a sample of studies reflects the limited resource we have to conduct the work.

We now clarify (page 6, lines 15-22) that two reviewers will pilot data extraction for 3 studies to identify any necessary changes in the data extraction template, and will then dual extract data from a further 10% of studies. If there are any significant discrepancies between the two reviewers, then we will conduct parallel data extraction for a further 10% of studies, again checking for discrepancies. This process will be repeated until no further discrepancies occur, at which stage the remaining data extraction will be completed by the single reviewer. At each stage a third author will be consulted when the two reviewers cannot resolve discrepancies.

C6. Risk of Bias: Again, two reviewers should independently assess risk of bias in every included study.

R6. As above, we have updated our protocol (end of page 6 & start of page 7) to state that the risk of bias tool will be piloted for 3 studies by the two reviewers, and then parallel assessment conducted for 10% of studies, checking discrepancies and repeating the process until no further discrepancies occur and a single reviewer will complete the process.

C7. GRADE quality appraisal: Again, two reviewers should independently assess study quality.

R7. We have updated the text of 'Confidence in cumulative evidence' section of the Methods to clarify that two reviewers will independently apply GRADE.

C8. Discussion: The discussion section is 4 lines long and needs revision. There is no discussion of the limitations or challenges of conducting this type of systematic review. Perhaps the authors could

recap on what previous reviews have found and state how they plan for theirs to be different and what specifically it can add to the evidence base.

R8. Thank you for these helpful suggestions. We had written a short discussion section as after reviewing other systematic review protocols published in the BMJ Open these either had no discussion section or a very short summary sentence. If the Editor is happy for a longer discussion, we have updated with your suggestion of a recap on what previous systematic reviews identified and how our review differs but builds on these previous findings. At the end of the Discussion (paragraph 4), again per your suggestion, we have added a limitations section.

C9. Page 4, line 14: The authors' last point is speculative as they have not conducted the review yet. Suggest removing.

R9. We have removed this limitation and replaced it with a limitation which note the applicability of our findings.

C10. Page 4, line 53: "...depends upon the ability of accurately identifying..." awkward sentence, suggest rewording. The ability of whom/what?

R10. Thank you for picking up on the issue with this phasing, we have reworded to "high quality EHR based research depends upon correct classification of cases and non-cases". We hope you will agree that this wording is clearer.

C11. Page 7, line 40: suggest including study quality as a subgroup analysis for heterogeneity.

R11. Thank you for this suggestion, we have updated the text to indicate that we will assess heterogeneity overall and in each subgroup with and without studies which have a high risk of bias (page 7, line 18).

Reviewer: 2

C12. For a small meta-analysis like this one, I'm not sure they need to publish a research protocol in BMJ. It seems a protocol for internal review is good enough.

R12. We appreciate the opinion that protocol publication is not necessary for small meta-analyses, but we feel that it is still worthwhile publishing our systematic review protocol in BMJ Open. Protocols are not only published for transparency purposes but also to avoid duplication. Research using EHR databases continues to expand, and with that the validity of data is often considered. We therefore wish to ensure that other researchers are aware of our work to avoid duplication. Indeed, the two existing systematic reviews which have validated stroke diagnoses in EHRs/administrative datasets were conducted at the same time presumably without being aware of each other's overlapping work since neither appear to have published protocols.

C13. They don't have clear statistical analysis plan for the outcomes in the protocol.

R13. We have edited and expanded the 'Data synthesis and meta-bias(es)' section (page 7) of our Methods section to provide further detail of our planned analysis, which we emphasise is dependent on the number of studies and the heterogeneity identified between studies.

VERSION 2 – REVIEW

REVIEWER	Dr. Niamh Merriman Royal College of Surgeons in Ireland Dublin, Ireland
REVIEW RETURNED	20-Sep-2019

GENERAL COMMENTS	The authors have addressed my comments. I have nothing further to suggest.
--